# Factors affecting the elderly's behavioral intention toward smart home systems: A cross-sectional study from China's eastern coast

**Yuan Wang** [1]*, **Norazmawati Md. Sani**[1], **Yinhong Hua**[2], **Qianling Jiang**[3], **Long Zhao**[2]

**1** School of Housing, Building and Planning, Universiti Sains Malaysia, Gelugor, Penang, Malaysia, **2** School of Art Design, Qufu Normal University, Rizhao, Shandong, China, **3** School of Design, Jiangnan University, Wuxi, Jiangsu, China

* yuanwang@student.usm.my

**Data Availability Statement:** All relevant data are available from the Figshare repository at the following URL: https://figshare.com/articles/dataset/Questionnaire_on_the_behavioral_

## Abstract

As China's population continues to age rapidly, smart elderly care has become a crucial strategy for addressing this societal challenge. Smart home systems have the potential to significantly enhance the quality of life, safety, and autonomy of the elderly. However, the adoption rate of these systems in this demographic remains relatively low, requiring an exploration of the key factors influencing the behavioral intention to use such systems. This study developed a comprehensive model based on the Technology Acceptance Model and the Unified Theory of Acceptance and Use of Technology. Through empirical analysis using partial least squares structural equation modeling, this study systematically examined the pathways through which various factors affected the behavioral intentions of the elderly. The findings revealed that perceived usefulness, facilitating conditions, compatibility, and perceived cost had significant direct effects on behavioral intention. Additionally, social influence, perceived trust, facilitating conditions, and compatibility indirectly influenced behavioral intention through perceived usefulness and perceived ease of use. Although perceived trust did not directly affect behavioral intention, it exerted an indirect influence through perceived usefulness. This comprehensive model provides theoretical insights into the behavioral intentions of the elderly toward smart home systems and offers practical guidance for developers, designers, and policymakers in the development, design, and promotion of these systems.

## Introduction

Since becoming an aging society in 2000, China's aging process has rapidly accelerated, with the country expected to enter a stage of severe aging by 2035 [1]. According to the "Bulletin of the Seventh National Population Census," China has the world's largest elderly population, particularly concentrated in the eastern coastal areas provinces, such as Shandong, Jiangsu, Guangdong, Hebei, and Zhejiang [2]. Fig 1A presents the top 10 provinces by elderly

intention_of_smart_home_systems_for_the_
elderly_/26763601?file=48621316.

**Funding:** The author(s) received no specific funding for this work.

**Competing interests:** The authors have declared that no competing interests exist.

population, and Fig 1B illustrates their geographical locations. This trend poses significant challenges across the social, economic, healthcare, and elderly care sectors. Socially and economically, the increasing proportion of the elderly has exacerbated labor market tensions, affecting productivity and economic growth [3]. In healthcare, the risk of chronic diseases among the elderly has increased sharply, leading to a substantial increase in healthcare demands [4]. Additionally, the high proportion of one-child families in the eastern region intensifies the burden of elderly care [5]. Age-friendly housing and community planning have become crucial [6]. These issues have garnered significant attention from governments and academia, leading to the introduction of multiple policies [7]. Over the past decade, China has increased its investment in social welfare, encouraging the participation of various social forces to gradually establish a multi-party system for providing elderly care. With advancements in the Internet of Things (IoT), information technology, big data, and cloud computing, the Chinese government has been actively promoting smart elderly care [8].

In China, "aging in place" remains the predominant model of elderly care. Statistics indicate that more than half of the elderly population in China lives in empty-nest households [1]. With age, older adults increasingly face physical decline and health challenges [9], leading to a growing need for information technology support [7]. Leveraging the IoT, smart home systems integrate household devices, appliances, and security features into an automated and remotely controllable environment [10]. These technologies offer numerous benefits to the aging process, for example, safety monitoring, emergency alerts, and anti-intrusion features of smart home systems significantly enhance the sense of security among elderly residents [11]. In addition, smart climate control, lighting systems, and appliances can automatically adjust to the preferences and behavioral patterns of the elderly, thereby improving their overall comfort [12]. Moreover, smart home systems empower older adults to manage daily activities independently, such as controlling devices via voice commands and remotely monitoring their health, thereby reducing their reliance on others [13].

Despite the significant improvements that smart home systems can offer to the elderly [14–17], their adoption rate in this demographic remains relatively low [18, 19]. Therefore, it

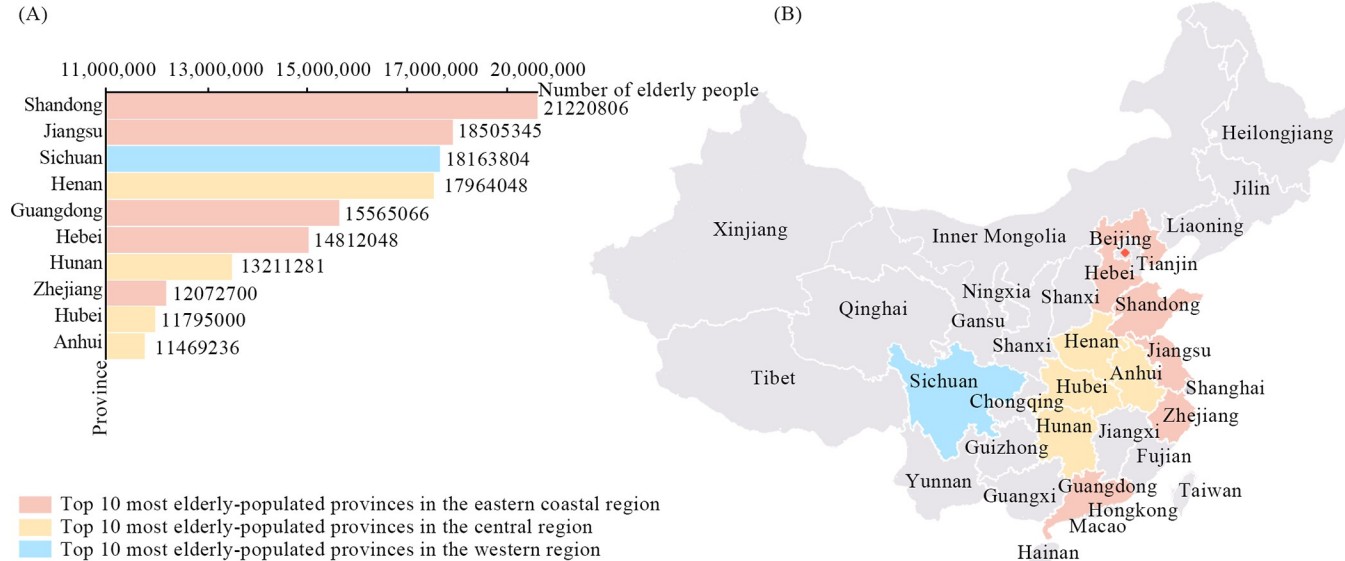

**Fig 1. The top 10 provinces in China with the largest elderly populations in 2020.** (A) Number of elderly people in the top 10 provinces, (B) The geographical locations of these provinces.

is essential to explore the key factors influencing the behavioral intentions of older adults. Although existing studies have extensively examined the technical aspects of smart home systems [11, 20, 21], there is a significant lack of empirical research from a user perspective, particularly in the Chinese context. This study aims to investigate the behavioral intentions of the elderly in the eastern coastal regions of China toward smart home systems. The findings of this research not only provide theoretical insights into the behavioral intentions of the elderly but also offer practical guidance for developers, designers, and policymakers in their efforts to innovate, design, and promote smart home systems.

The structure of this paper is as follows: The first section introduces the research background, research questions, and objectives. The second section provides a literature review and proposes research hypotheses. The third section details the research methodology, including the processes, methods, and tools used. The fourth section presents the data analysis and results, in which the hypotheses are tested using structural equation modeling. The fifth section discusses the research findings. Finally, the conclusion summarizes the key findings and provides suggestions for future research.

## Literature review

### Smart home systems for the elderly

Smart home systems leverage the IoT technology to connect devices and appliances, thereby enabling automated management and remote control [10]. In recent years, researchers have extensively explored the benefits of smart home systems for elderly populations. These studies primarily focused on advancements in health monitoring, voice control, and behavior prediction from the perspectives of technology and design methods. From a technical standpoint, Zou et al. [21] designed a Raspberry Pi-based smart home system to detect fatigue levels in older adults and control appliances. Other scholars have examined the application of voice-control technology in smart home systems to enhance their ease of use by older users [22]. From a design perspective, Cui and Chung [23] analyzed the design methods for smart elderly care housing under the concept of "user-centered" design, emphasizing the importance of meeting user needs. Yun et al. [24] proposed an information fusion model for designing age-appropriate home environments and developed a behavioral prediction system for the elderly. Additionally, researchers have investigated the perceptions of smart home systems among the elderly in South Korea [25]. These studies have primarily focused on technical aspects, with limited research from a user perspective, particularly in the context of China.

### Theoretical background

Researchers have applied various theoretical models to studies related to smart homes for the elderly, such as the Technology Acceptance Model (TAM) [7, 26–28], Unified Theory of Acceptance and Use of Technology (UTAUT) [29–31], Theory of Planned Behavior [32], and Theory of Reasoned Action (TRA) [32]. These models aim to explain the factors driving individuals to adopt technology, with the TAM and UTAUT being the most widely used. However, most studies have focused on the use of smart home technologies and devices by the elderly, and there is a lack of research on smart home systems based on these theories. To address this gap, this section constructs a comprehensive model based on the TAM, UTAUT, and extended variables.

**Technology Acceptance Model (TAM).** The TAM is a highly reliable and empirically supported theoretical model [33]. Derived from TRA, the TAM focuses on understanding users' behavioral intentions to voluntarily use information technology systems [34]. In the TAM, perceived ease of use and perceived usefulness are core variables for predicting users'

behavioral intentions [34]. In the early version of the TAM, attitude was considered a mediating variable between perceived ease of use, perceived usefulness, and behavioral intention. However, as the model evolved, attitude was found to have an insignificant predictive effect on behavioral intention and was therefore removed in subsequent studies [35, 36]. Today, the TAM is recognized as one of the most popular and persuasive models for studying user acceptance of information technology [7, 33].

The TAM has been widely applied across various fields including education [37, 38], business [39–41], computer science and information systems [42–44], and green sustainable technology [45, 46]. In the context of smart home systems for the elderly, Pal et al. [47] examined the factors influencing the adoption of smart home services by the elderly using the TAM. Etemad-Sajadi and Gomes Dos Santos [28] explored the use of IoT health technologies in the homes of older adults guided by the TAM. Yan & Lee [27] developed a model for a smart home healthcare system for the elderly based on the TAM. These studies demonstrate that perceived usefulness and perceived ease of use have strong explanatory power for the elderly's adoption of smart home technologies and devices. Although multiple studies indicate that the TAM is applicable in the context of smart homes for the elderly, scholars have pointed out that it overlooks the influence of social factors on the technology adoption process [33]. This limitation could be overcome by incorporating external variables.

**Unified Theory of Acceptance and Use of Technology (UTAUT).**   The UTAUT is derived from the integration of eight theories: the Theory of Planned Behavior (TPB), TAM, combination of TAM and TPB, TRA, Model of Personal Computer Utilization, Diffusion of Innovation Theory, Social Cognitive Theory and Motivation Model [48]. The UTAUT model comprises four key constructs: social influence, facilitating conditions, performance expectancy, and effort expectancy. Notably, the UTAUT includes social influence, thus addressing the limitations of the TAM, which tends to overlook social factors.

The UTAUT has been applied in various fields, such as healthcare [49], smart technology [30, 31], mobile commerce [39], higher education [49], and electric vehicles [50]. The UTAUT has also been widely used in studies related to smart homes for older adults. Scholars have pointed out that the UTAUT model was initially developed based on systems in workplace environments, in which the determining factors may differ from those in a home setting [33]. However, many empirical studies on UTAUT have been conducted in home environments [29–32, 47, 51–53]. Arar et al. [31] used the UTAUT framework to analyze the acceptance and preferences of older adults in Dubai, UAE, toward smart home technology. Maswadi et al. [30] employed the UTAUT model to study the behavioral intentions of the elderly in Saudi Arabia to adopt smart home technologies. Zhong et al. [29] used the UTAUT framework to investigate differences in the acceptance of smart home voice assistants across age groups. These studies largely identified social influence and facilitating conditions as key variables and validated their effectiveness.

**A comparison between the TAM and UTAUT.**   A comparison between the TAM and UTAUT shows that the perceived usefulness in the TAM corresponds to performance expectancy in the UTAUT, whereas perceived ease of use corresponds to effort expectancy [30, 54]. Perceived usefulness and perceived ease of use were selected as the research variables for two main reasons. First, compared with performance expectancy and effort expectancy, perceived usefulness and perceived ease of use have been widely applied in studies on the acceptance of smart home technologies, confirming their applicability in the smart home domain [7, 13, 26, 27]. Second, multiple studies have confirmed the mediating effects of perceived usefulness and perceived ease of use on other research variables and behavioral intention [55–57]. The mediating roles of perceived usefulness and perceived ease of use allow for a more comprehensive understanding of the complex mechanisms underlying the behavioral intentions of the elderly.

Several scholars have integrated variables from the TAM and UTAUT for empirical research [35, 36, 53]. Researchers have combined variables such as social influence, facilitating conditions, perceived usefulness, perceived ease of use, and compatibility to construct a comprehensive model examining older adults' behavioral intentions toward smart wearable devices [53]. The results indicated that perceived usefulness and perceived ease of use mediated the relationship between various factors and behavioral intention, thus enhancing the understanding of user behavioral intentions.

**Extended variables of the TAM and UTAUT.**   In the field of smart homes, several studies have incorporated perceived cost, perceived trust, and compatibility as extended variables in the TAM and UTAUT. These studies have demonstrated that these variables have a direct or indirect effect on behavioral intention [13, 55, 57–60]. Al-Bashayreh et al. [55] noted that compatibility indirectly influences behavioral intention by affecting perceived usefulness and perceived ease of use. Dhagarra et al. [57] found that perceived trust indirectly affects behavioral intention through perceived usefulness. Abu-Taieh et al. [59] proposed that perceived cost has a direct effect on behavioral intention. Therefore, this study incorporated perceived cost, perceived trust, and compatibility as extended variables in the research model.

Despite the widespread application of the TAM and UTAUT in studies on smart home technologies and devices, research on elderly people's behavioral intention toward smart home systems based on these theoretical models remains limited. Building on the TAM, UTAUT, and their extended variables, this study proposed a comprehensive theoretical framework (Fig 2).

## Hypotheses development

**Variables of the TAM.**   *Perceived ease of use*. As an important variable in the TAM, perceived ease of use refers to the degree to which users find a specific technology or system easy to use [61]. In this study, perceived ease of use is described as the elderly user's perception of the ease of use of smart home systems. Previous studies indicate that perceived ease of use significantly influences perceived usefulness. Song et al. [13] found that perceived ease of use positively affects perceived usefulness in the adoption of voice user interfaces by older adults. Wei et al. [26] and Zhou et al. [7] corroborate this finding. Therefore, the following hypothesis is proposed:

H1: Perceived ease of use positively affects perceived usefulness.

*Behavioral intention*. Behavioral intention is a key variable in the TAM, reflecting a user's subjective likelihood of using a particular technology or system in the future and serves as an important predictor of user behavior [61]. In studying older adults' behavioral intentions toward smart home healthcare systems, Yan and Lee [27] found that perceived ease of use significantly influences elderly users' behavioral intentions. Similarly, Wei et al. [26] observed that perceived ease of use positively affects behavioral intention in the context of older adults' willingness to use smart homes. Hence, the following hypothesis is proposed:

H2: Perceived ease of use positively affects behavioral intention.

*Perceived usefulness*. In the TAM, perceived usefulness refers to the extent to which users believe that using a particular technology or system improves their job performance [61]. In this study, perceived usefulness is defined as the extent to which older adults believe that using smart home systems will positively influence their quality of life and daily activities. Previous studies demonstrated that elderly users generally perceive the usefulness of smart home healthcare systems as a significant factor influencing their behavioral intentions [27]. Zhou et al. [7]

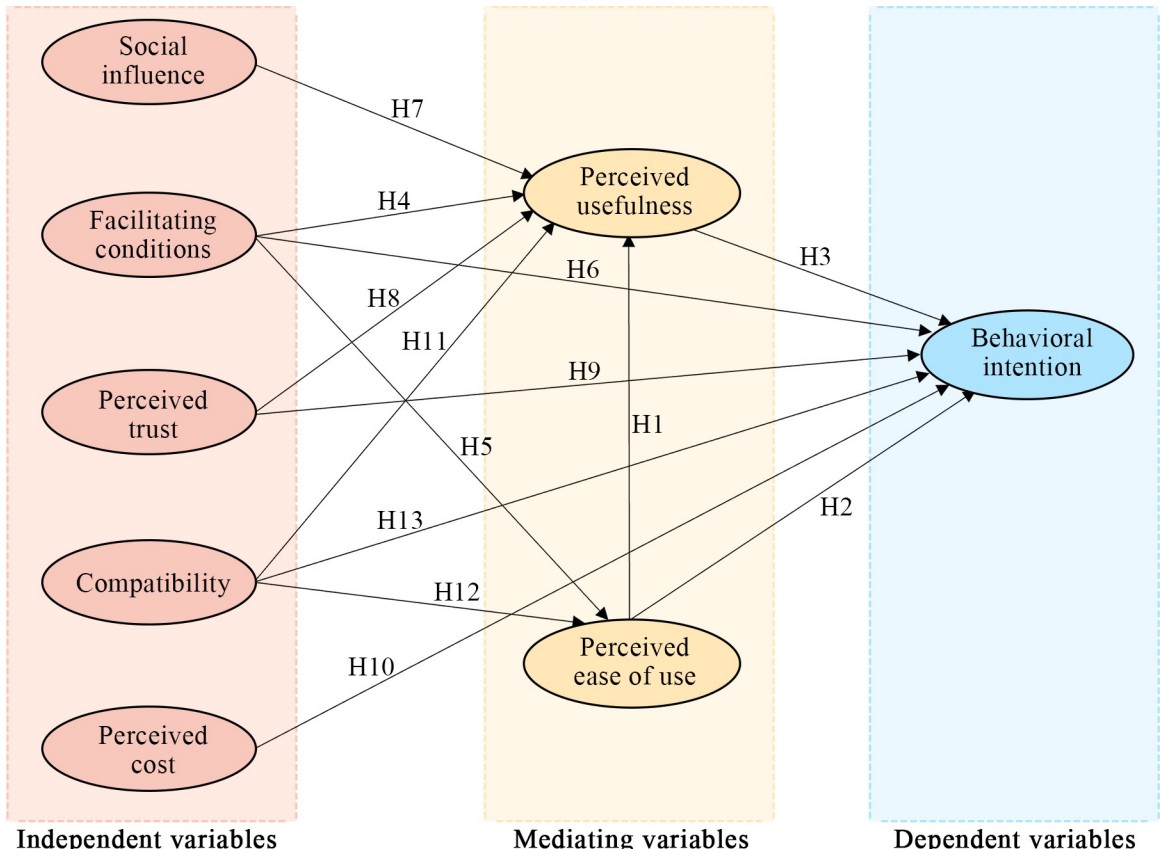

**Fig 2. Theoretical framework. Note: Red** – Social influence and facilitating conditions are fundamental constructs in the UTAUT, serving as independent variables. Perceived trust, compatibility, and perceived cost are extended constructs that function as independent variables. **Yellow** –Perceived usefulness and perceived ease of use are fundamental constructs in the TAM, acting as mediating variables. **Blue** – Behavioral intention is a key construct in both the TAM and UTAUT, serving as the dependent variable.

confirmed the positive effect of perceived usefulness on behavioral intention in their study of elderly consumers' willingness to adopt smart homes. Based on these findings, the following hypothesis is proposed:

H3: Perceived usefulness positively affects behavioral intention.

**Variables of the UTAUT.** *Facilitating conditions.* In the UTAUT, facilitating conditions refer to the degree to which individuals believe that external resources and the environment are available to support their use of the system [48, 62]. In this study, facilitating conditions refer to whether older adults have access to the necessary support, resources, and an appropriate environment to help them use smart home systems smoothly and effectively. In a study on e-learning adoption, scholars recognized facilitating conditions as significant determinants of perceived usefulness [63]. Chen and Aklikokou's [64] study on e-government adoption also supports this relationship. Therefore, the following hypothesis is proposed:

H4: Facilitating conditions positively affect perceived usefulness.

Furthermore, previous studies have demonstrated that facilitating conditions significantly influence perceived ease of use. Technical training and support can reduce usage difficulties,

making it easier for users to perceive a system as user-friendly. Sukendro et al. [63] confirmed the positive effect of facilitating conditions on perceived ease of use using an extended TAM. George et al. [65] noted that facilitating conditions play a crucial role in enhancing perceived ease of use. Similarly, the study by Chen and Aklikokou [64] supports this finding. Hence, the following hypothesis is proposed:

H5: Facilitating conditions positively affect perceived ease of use.

In addition, previous studies found that facilitating conditions had a significant positive effect on behavioral intention. Providing the elderly with appropriate technical training and support can reduce the difficulty of using technology, thereby enhancing their willingness to do so. Zhong et al. [29] investigated the acceptance of smart home voice assistants and found that facilitating conditions enhanced behavioral intentions of elderly users [30]. Similarly, the study by Jahanshahi et al. [66] confirmed the positive relationship between facilitating conditions and behavioral intention. Based on this analysis, the following hypothesis is proposed:

H6: Facilitating conditions positively affect behavioral intention.

*Social influence*. Social influence is a key variable for understanding users' technology acceptance behavior [48]. In the UTAUT, social influence refers to the extent to which individuals perceive that others in their social group believe that they should use a new technology or system [48]. In this study, social influence is defined as the extent to which users perceive that the people in their group explicitly encourage and approve of their use of smart home systems. Previous studies have shown that social influence positively affects perceived usefulness. In a study of Jordanian users' acceptance of IoT health systems, Alkhwaldi and Abdulmuhsin [44] identified social influence as a significant determinant of perceived usefulness. Zhang et al. [56] confirmed this finding in their study on the acceptance of autonomous vehicles in China. Accordingly, the following hypothesis is proposed:

H7: Social influence positively affects perceived usefulness.

**Extended variables.** *Perceived trust*. Smart home systems collect data on users and their home environments through sensors, which may raise concerns about personal privacy and consequently affect trust in the system [67]. Perceived trust refers to the psychological state of older adults who believe that their personal data are securely protected, handled with care, and kept anonymous [67]. Talwar et al. [68] and Dhagarra et al. [57] demonstrate that perceived trust significantly enhances perceived usefulness, indicating that trust plays a crucial role in technology acceptance. Alshurafat et al. [69] further emphasize that trust can reduce uncertainty and significantly increase perceived usefulness. Therefore, the following hypothesis is proposed:

H8: Perceived trust positively affects perceived usefulness.

Moreover, perceived trust was found to significantly affect users' behavioral intentions. Song et al. [13] identified perceived trust as a key factor influencing the acceptance of voice user interfaces in smart home systems among the elderly in China. Similarly, Inder et al. [70] highlight in their study on online banking adoption that perceived trust exerts a significant positive effect on behavioral intention. This finding is further confirmed by Sujood et al. [71]. Hence, the following hypothesis is proposed:

H9: Perceived trust positively affects behavioral intention.

*Perceived cost*. Perceived cost is a key factor that influences technology adoption [72]. Perceived cost encompasses various costs and expenses that users perceive when deciding to adopt a product, service, or technology [67]. In this study, perceived cost refers to the users' subjective perceptions and understanding of the costs and expenses associated with using and maintaining smart home systems, including direct economic, time, and learning costs. In the context of mobile shopping, scholars have noted that having higher perceived cost significantly reduces users' behavioral intentions [59]. Similarly, Arfi et al. [73] found that perceived cost significantly hindered acceptance of e-health services. Based on these insights, the following hypothesis is proposed:

H10: Perceived cost negatively affects behavioral intention.

*Compatibility*. Compatibility is a key variable in technology adoption [74]. Compatibility describes the degree to which an innovation aligns with the existing beliefs, habits, and experiences of end users [53, 75]. In the context of smart home systems for the elderly, compatibility pertains to a system's compatibility with the users' existing habits, experiences, and smart devices. Yan and Lee [27] noted in their study on smart home healthcare systems that compatibility is a significant factor that influences perceived usefulness. When a system aligns with the living habits and technological experiences of older adults, users are more likely to recognize the value and benefits of technology. Thus, the following hypothesis is proposed:

H11: Compatibility positively affects perceived usefulness.

Moreover, studies indicate that compatibility has a significant positive effect on perceived ease of use. In the mobile payments field, scholars have observed that Alipay's high compatibility with consumers' payment habits and financial systems effectively enhances users' perceived ease of use [60]. Similarly, Al-Bashayreh et al. [55] and Ullah et al. [74] demonstrated that, when the design of technology or systems aligned with user habits, experiences, and existing devices, resistance to use diminished, leading to a significant increase in perceived ease of use. Based on this, the following hypothesis is proposed:

H12: Compatibility positively affects perceived ease of use.

Compatibility was also found to have a significant positive effect on behavioral intention. Previous studies suggest that higher compatibility enhances user acceptance of new technologies. Octavius and Antonio [76] found that compatibility significantly increased behavioral intention in a study on the adoption of mobile health applications. Similarly, in their research on mobile shopping applications, Rattanaburi and Vongurai [59] observed a significantly positive effect of compatibility on behavioral intention. Fazal-e-Hasan et al. [77] further confirmed this positive relationship. Based on these findings, the following hypothesis is proposed:

H13: Compatibility positively affects behavioral intention.

## Research methodology

This study used structural equation modeling (SEM) for quantitative research. As a multivariate analysis tool, SEM is widely used across various fields because of its ability to comprehensively examine complex relationships between variables [78]. Researchers in this field have adopted SEM for their analyses [7, 26, 29, 30]. As this study aimed to validate the relationships between multiple independent and dependent variables, this method was considered appropriate. SEM includes models based on partial least squares (PLS-SEM) and covariance-based (CB-SEM) [78]. PLS-SEM typically employs Smart PLS software for data analysis, whereas CB-SEM is conducted using Amos.

## PLS-SEM

This study used PLS-SEM for data analysis based on three key considerations. First, compared with CB-SEM, PLS-SEM is more suitable for non-normally distributed data. Because data in this study did not follow a normal distribution, the selection of PLS-SEM ensured the robustness of the analysis and predictive capability of the model [79]. Second, PLS-SEM emphasizes the predictive power and explanatory ability of the model, making it particularly well-suited for models involving multiple variables and complex relationships [80]. As the model in this study involved complex relationships between several variables, PLS-SEM was considered more appropriate. Additionally, PLS-SEM has been widely applied in this field, providing a reference and guidance for our methodological choice [26, 27, 30].

## Survey instrument

This study used an online questionnaire as a data collection tool based on prior studies in this field [13, 27, 29, 30, 81]. The validity of this method has been well- established, as it is the most widely used survey method in this area of research. The questionnaire was divided into two parts. The first part collected demographic information, and the second part contained measurement items for the research variables, which were adapted from previous studies and appropriately modified to fit the specific context of this research. All measurement items and their corresponding literature sources are listed in Table 1. A 7-point Likert scale (1 = "strongly disagree"; 7 = "strongly agree") was used to numerically analyze the relationships between variables.

## Pilot study

A pilot study and expert validation were conducted before collecting data [30]. In this pilot study, 35 participants from diverse backgrounds were invited to evaluate the reliability and representativeness of the questionnaire. After completing the questionnaire, the respondents provided feedback on content relevance, language clarity, and comprehension difficulty. In addition, three experts in smart home technology were consulted to review the questionnaires. They provided in-depth suggestions regarding the structure and content, including the accuracy of wording, professionalism of the language, and optimization of the question order. Based on the feedback from both experts and participants, comprehensive adjustments were made to the questionnaire to ensure its suitability for subsequent data collection.

## Respondents

The respondents in this study were individuals aged 55 years and older, residing in the eastern coastal regions of China. The age range was selected based on the standards of previous studies [35, 36, 67, 85]. To enhance the validity of the data, respondents were required to have knowledge of or experience with smart home systems [27]. All respondents were fully informed about the purpose of the study and their roles before participating, ensuring that they provided informed consent and accurately understood the background and objectives of the study.

## Sample size

The sample size for this study was calculated using Cochran's formula, which was specifically designed for research involving large populations samples [86]. This method ensures an

**Table 1. Measurement items.**

| Construct | Item | Question (responses based on 7 points Likert scale) | References |
|---|---|---|---|
| Perceived usefulness | PU 1 | I will accomplish my daily activities more quickly by using smart home systems. | [82–84] |
| | PU 2 | Using smart home systems will enhance my overall quality of life. | |
| | PU 3 | Using smart home systems will make my life more convenient. | |
| | PU 4 | Using smart home systems will make my life more useful. | |
| | PU 5 | Using smart home systems will make my life more enjoyable. | |
| Perceived ease of use | PEOU 1 | I will operate smart home systems in my home by myself. | [35, 47] |
| | PEOU 2 | I will find smart home systems easy to use. | |
| | PEOU 3 | Using smart home systems will not require any special mental or physical effort. | |
| | PEOU 4 | It will be easy and clear for me to use the smart home system installed in my home. | |
| | PEOU 5 | I will be skillful at using smart home systems. | |
| Social influence | SI 1 | I will use smart home systems if the media/government encourages me to use them. | [32, 67] |
| | SI 2 | I will use smart home systems in my house if my family members and friends do so. | |
| | SI 3 | I will use smart home systems if people whose opinions I value recommend that I do so. | |
| | SI 4 | People who are important to me will support my use of smart home systems. | |
| Facilitating conditions | FC 1 | It is important to have someone who can help me tackle problems in the use of smart home systems. | [35, 67] |
| | FC 2 | Training and practice are useful and important for the use of smart home systems. | |
| | FC 3 | I believe proper guidance will be available when using smart home systems. | |
| | FC 4 | I believe proper service will be available if I face difficulties in using smart home systems. | |
| Perceived trust | PT 1 | I fear using smart home systems due to the potential loss of my personal data and privacy. | [67] |
| | PT 2 | The internet offers a medium through which sensitive personal information can be sent confidentially. | |
| | PT 3 | I think it is risky to disclose my personal details and health information to smart home service providers. | |
| Perceived cost | PC 1 | The daily cost of smart home systems should be economical. | [35, 67] |
| | PC 2 | I will need to pay a much lower price for traditional home devices than for subscribing to smart home systems. | |
| | PC 3 | Purchasing and maintaining smart home systems will be a burden for me. | |
| | PC 4 | The cost of investing in smart home systems is too expensive. | |
| Compatibility | COM 1 | Smart home systems can be compatible with my existing electronics (smartphones and other devices). | [47, 53] |
| | COM 2 | Purchasing smart home devices from different vendors will not create any operational problems. | |
| | COM 3 | The smart home devices can inter-operate with each other. | |
| | COM 4 | Using smart home systems will fit into all aspects of my work. | |
| | COM 5 | Using smart home systems will not affect my daily life. | |
| Behavioral intention | BI 1 | I will be interested in smart home systems. | [32, 53, 67] |
| | BI 2 | Using smart home systems is a good idea. | |
| | BI 3 | I expect to use smart home systems in my house. | |
| | BI 4 | I will use smart home systems in the future. | |
| | BI 5 | I intend to invest in and use smart home systems as much as possible. | |

optimal sample size at a predetermined precision level. The formula is as follows:

$$n_{55} = \frac{z^2 . p . (1-p)}{e^2}$$

Where:
$n$ denotes the sample size
$e$ denotes the allowable error margin of error
$p$ is the estimated proportion of the target characteristic
$z$ is the z-value corresponding to the standard normal distribution.

**Table 2. Sample size of the elderly based on region.**

| Region | Number of the elderly | Proportion of the elderly population (%) | sample size |
|---|---|---|---|
| Shandong | 21220806 | 18.4 | 102 |
| Jiangsu | 18505345 | 16 | 89 |
| Guangdong | 15565066 | 13.5 | 75 |
| Hebei | 14812048 | 12.8 | 65 |
| Zhejiang | 12072700 | 10.5 | 58 |
| Liaoning | 10954467 | 9.5 | 53 |
| Guangxi | 8363779 | 7.2 | 40 |
| Fujian | 6637869 | 5.8 | 32 |
| Shanghai | 5815462 | 5.0 | 28 |
| Hainan | 1476599 | 1.3 | 8 |

Based on the parameter set for this study ($z=1.96$, $p=0.5$, and $e=0.05$), the calculation was as follows:

$$n_{55} = \frac{1.96^2 . 0.5 . (1-0.5)}{0.05^2} = 384.16 \approx 385$$

To account for the expected 70% response rate [87], the study included 550 respondents.

$$\frac{385}{0.70} \approx 550$$

## Sampling procedure

Sampling involves selecting a representative subset of units from the overall population [88]. The stratified sampling method divides the population into mutually exclusive strata and then randomly selects samples from each stratum rather than sampling directly from the entire population. This method is advantageous over simple random sampling because it reduces within-stratum heterogeneity, resulting in a more representative sample [89]. In this field, scholars have recommended stratified sampling to enhance representativeness [90]. This study selected China's eastern coastal regions, which have the largest aging populations, as the survey location. Stratified sampling was conducted based on the proportion of the elderly population in each province to minimize bias owing to regional differences. The sampling results are presented in Table 2.

## Inclusivity in global research

Additional information regarding the ethical, cultural, and scientific considerations specific to inclusivity in global research is included in the Supporting Information (S1 File).

## Ethical statement

This study received ethical approval from the Biomedical Ethics Committee of Qufu Normal University (Approval number: 2024051).

## Informed consent

In this study, informed consent was obtained orally from all participants. Written consent was not feasible due to the nature of the online survey, which was conducted remotely without

direct interaction between the researchers and participants. The Biomedical Ethics Committee of Qufu Normal University approved the use of oral consent for this study. Informed consent was obtained directly from respondents through "WJX", an online questionnaire platform. Prior to completing the questionnaire, respondents were required to review and agree to a detailed informed consent form. This form clearly described the purpose of the study, the voluntary nature of participation, confidentiality of their responses, and respondents' rights. The research team provided detailed explanations of how the respondents' privacy would be protected, ensuring data anonymity and security. Respondents were informed that their participation was entirely voluntary and that they could withdraw from the study at any time without providing a reason. All collected data would be kept strictly confidential and used solely for research purposes. Only after respondents had thoroughly reviewed and understood the content of the informed consent form and explicitly agreed by clicking the "agree" button were they permitted to proceed with the questionnaire.

## Data collection

Data for this study were collected between March 29, and April 12, 2024 using "WJX", a well-established and professional online data collection platform in China. In the questionnaire, the researchers explained the basic concepts of smart home systems to the respondents and provided detailed instructions to help them understand the study's context and questions. In total, 550 questionnaires were collected. The researchers cleaned the data and removed responses with excessively short completion times, incomplete answers, and uniform responses. This process yields 475 valid responses. According to Hair et al. [91], the sample size should be at least 10 times the number of scale items. Therefore, 475 samples were sufficient to support the testing and analysis using SEM.

## Data analysis and results

The data analysis was divided into two parts. The first part involved conducting a descriptive statistical analysis using SPSS 24.0, whereas the second part employed SEM using Smart PLS 4.0. This included an evaluation of both the measurement and structural models. The measurement model analysis focused on verifying the reliability and validity of the variables and indicators, whereas the structural model evaluation was used to test the proposed hypotheses [30].

### Demographic information

Table 3 presents respondents' demographic information. The majority were aged 55–65 years (66.8%), with a relatively balanced sex ratio. Most respondents lived with their children (67.2%), and their educational level was predominantly high school or above (52.8%). Regarding work status, nearly half of the respondents were retired (48.6%), and their income was primarily derived from alimony or government subsidies (41.5%). In terms of occupation, a significant proportion were employed by privately owned enterprises (40.0%), followed by liberal professions (25.7%) and employees of foreign capital enterprises (24.4%).

### SEM analysis

**Data analysis for the measurement model.** *Mitigation of positional bias*. Position bias refers to a phenomenon in which user behavior and selection outcomes are influenced by the positions of items or options on a page or list [92]. To address this issue, a small-scale pilot study was conducted by randomizing the order of the questions and options in the

**Table 3. Demographic information of respondents.**

| Sample | Category | Frequency | Percentage |
|---|---|---|---|
| Age | 55-60 | 138 | 29.1% |
| | 61-65 | 179 | 37.7% |
| | 66-70 | 95 | 20.0% |
| | >70 | 63 | 13.3% |
| Gender | Male | 233 | 49.1% |
| | Female | 242 | 50.9% |
| Living status | Not living with children | 156 | 32.8% |
| | Living with children | 319 | 67.2% |
| Education | Junior high school or below | 224 | 47.2% |
| | High school or above | 251 | 52.8% |
| Work status | Full-time work | 102 | 21.5% |
| | Part-time work | 115 | 24.2% |
| | Retired | 231 | 48.6% |
| | Never worked | 27 | 5.7% |
| Primary means of living | Salary/wages | 127 | 26.7% |
| | Property income | 151 | 31.8% |
| | Alimony and/or government subsidies | 197 | 41.5% |
| Occupation | Privately owned enterprises | 190 | 40.0% |
| | Foreign capital enterprises | 116 | 24.4% |
| | Public sector or state-owned enterprises | 47 | 9.9% |
| | Liberal professions | 122 | 25.7% |

questionnaire to assess potential position bias. The order of the questions and scale items was adjusted based on participant feedback and expert recommendations, and neutral wording was used to minimize language bias. These steps were taken to reduce the impact of position bias on the study results.

*Common method bias (CMB)*. Survey research is often susceptible to common method bias (CMB) [93], which requires careful control. Although previous researchers have used Harman's single-factor test to detect CMB, the reliability of this method has been questioned [30]. Consequently, scholars recommend using the variance inflation factor (VIF) to detect full collinearity with the commonly accepted threshold of 3.33 [30, 80, 94]. A VIF test was conducted to ensure data accuracy. Table 4 shows that the VIF values in this study range from 1.216 to 1.697, all of which are below the specified threshold [95], confirming that there are no CMB issues in the survey data.

**Table 4. Full collinearity.**

| | BI | COM | FC | PC | PEOU | PT | PU | SI |
|---|---|---|---|---|---|---|---|---|
| Behavioral intention | | | | | | | | |
| Compatibility | 1.617 | | | | 1.216 | | 1.585 | |
| Facilitating conditions | 1.447 | | | | 1.216 | | 1.476 | |
| Perceived cost | 1.552 | | | | | | | |
| Perceived ease of use | 1.697 | | | | | | 1.673 | |
| Perceived trust | 1.577 | | | | | | 1.500 | |
| Perceived usefulness | 1.562 | | | | | | | |
| Social influence | | | | | | | 1.600 | |

**Table 5. Standardized factor loadings, Cronbach's alphas, CRs, and AVEs.**

| Construct | Item | Factor loading | Cronbach's alpha | rho_A | Composite reliability | AVE |
|---|---|---|---|---|---|---|
| Perceived usefulness | PU 1 | 0.855 | 0.912 | 0.914 | 0.935 | 0.741 |
| | PU 2 | 0.855 | | | | |
| | PU 3 | 0.854 | | | | |
| | PU 4 | 0.874 | | | | |
| | PU 5 | 0.865 | | | | |
| Perceived ease of use | PEOU 1 | 0.842 | 0.901 | 0.901 | 0.926 | 0.716 |
| | PEOU 2 | 0.838 | | | | |
| | PEOU 3 | 0.863 | | | | |
| | PEOU 4 | 0.848 | | | | |
| | PEOU 5 | 0.838 | | | | |
| Social influence | SI 1 | 0.876 | 0.895 | 0.896 | 0.927 | 0.761 |
| | SI 2 | 0.866 | | | | |
| | SI 3 | 0.873 | | | | |
| | SI 4 | 0.875 | | | | |
| Facilitating conditions | FC 1 | 0.878 | 0.888 | 0.891 | 0.922 | 0.748 |
| | FC 2 | 0.850 | | | | |
| | FC 3 | 0.861 | | | | |
| | FC 4 | 0.871 | | | | |
| Perceived trust | PT 1 | 0.879 | 0.860 | 0.860 | 0.914 | 0.781 |
| | PT 2 | 0.887 | | | | |
| | PT 3 | 0.885 | | | | |
| Perceived cost | PC 1 | 0.870 | 0.896 | 0.897 | 0.928 | 0.762 |
| | PC 2 | 0.876 | | | | |
| | PC 3 | 0.886 | | | | |
| | PC 4 | 0.860 | | | | |
| Compatibility | COM 1 | 0.855 | 0.909 | 0.911 | 0.932 | 0.734 |
| | COM 2 | 0.857 | | | | |
| | COM 3 | 0.859 | | | | |
| | COM 4 | 0.870 | | | | |
| | COM 5 | 0.842 | | | | |
| Behavioral intention | BI 1 | 0.838 | 0.908 | 0.909 | 0.931 | 0.731 |
| | BI 2 | 0.850 | | | | |
| | BI 3 | 0.857 | | | | |
| | BI 4 | 0.870 | | | | |
| | BI 5 | 0.860 | | | | |

*Reliability and validity test.* The reliability and validity of the data were assessed using the Smart PLS software (version 4.0). Results in Table 5 show that Cronbach's alpha for all constructs exceeded 0.7, indicating a high reliability of the questionnaire and providing a solid data foundation for further analysis [96]. Additionally, the average variance extracted (AVE) for each construct was above 0.5, demonstrating the convergent validity of the measurement model [97].

In accordance with previous research, the Heterotrait-monotrait (HTMT) ratio should remain below 0.90 [26, 98]. As shown in Table 6, the results confirmed that these constructs satisfied the criteria for discriminant validity.

**Table 6. Heterotrait-monotrait ratio (HTMT).**

| Construct | BI | COM | FC | PC | PEOU | PT | PU | SI |
|---|---|---|---|---|---|---|---|---|
| Behavioral intention | | | | | | | | |
| Compatibility | 0.534 | | | | | | | |
| Facilitating conditions | 0.525 | 0.468 | | | | | | |
| Perceived cost | 0.546 | 0.509 | 0.444 | | | | | |
| Perceived ease of use | 0.511 | 0.561 | 0.526 | 0.519 | | | | |
| Perceived trust | 0.446 | 0.521 | 0.444 | 0.463 | 0.531 | | | |
| Perceived usefulness | 0.513 | 0.453 | 0.430 | 0.516 | 0.471 | 0.550 | | |
| Social influence | 0.504 | 0.517 | 0.524 | 0.503 | 0.540 | 0.525 | 0.489 | |

In addition, we tested discriminant validity, as shown in Table 7. Notably, the diagonal elements (i.e., the square roots of the AVE) indicated higher correlations between the constructs. Thus, most constructs in this study demonstrated good discriminant validity [99].

**Data analysis for the structural model.** *Model fit test.* This study used Smart PLS 4.0 to test the hypotheses and validate the model. Previous research suggests an $R^2$ value of at least 0.26 is considered reasonable, whereas in exploratory studies, an $R^2$ value exceeding 0.20 is acceptable [80]. The results of this study indicate that the $R^2$ values for behavioral intention, perceived usefulness, and perceived ease of use exceed 0.20, indicating that the model adequately explains the variability of these variables. Scholars assert that a $Q^2$ value greater than zero confirms the predictive relevance of a model [80, 95]. The $Q^2$ values in this study met these standards, confirming the predictive relevance of the model. Table 8 shows the detailed model fit indices.

Additionally, scholars have proposed methods for calculating the goodness of fit (GoF) [26, 100]. The GoF value is calculated as follows:

$$GoF = \sqrt{\overline{AVE} \times \overline{R^2}} = \sqrt{0.747 \times 0.365} \approx 0.522$$

Based on this calculation, GoF was 0.522. This value exceeds the thresholds suggested by Tenenhaus et al. [100] and Wei et al. [26], which are greater than 0.36, further indicating that the model demonstrates a good overall fit.

*Hypothesis testing analysis.* The hypotheses were tested by bootstrapping using Smart PLS 4.0 [101]. Previous scholars have indicated that an SRMR value of less than 0.08 is acceptable. The SRMR value obtained in this study was 0.058 [26], which met the specified standard.

The results of the model path effects are presented in Table 9 and Fig 3. Except for H9 (the impact of perceived trust on behavioral intention), all paths in the model are statistically

**Table 7. Correlation matrix among the constructs and square root of AVEs.**

| Construct | BI | COM | FC | PC | PEOU | PT | PU | SI |
|---|---|---|---|---|---|---|---|---|
| Behavioral intention | **0.855** | | | | | | | |
| Compatibility | 0.487 | **0.857** | | | | | | |
| Facilitating conditions | 0.473 | 0.421 | **0.865** | | | | | |
| Perceived cost | -0.495 | -0.460 | -0.396 | **0.873** | | | | |
| Perceived ease of use | 0.463 | 0.510 | 0.471 | -0.466 | **0.846** | | | |
| Perceived trust | 0.396 | 0.462 | 0.389 | -0.406 | 0.468 | **0.884** | | |
| Perceived usefulness | 0.469 | 0.414 | 0.389 | -0.469 | 0.428 | 0.488 | **0.861** | |
| Social influence | 0.455 | 0.467 | 0.467 | -0.452 | 0.486 | 0.461 | 0.444 | **0.872** |

**Table 8. Model fit indices.**

| Construct | $R^2$ | $Q^2$ |
|---|---|---|
| Behavioral intention | 0.415 | 0.298 |
| Perceived ease of use | 0.340 | 0.239 |
| Perceived usefulness | 0.340 | 0.247 |

significant. This underscores the model's significant contribution in explaining older adults' behavioral intentions toward smart home systems.

## Discussion

This section provides an in-depth discussion of the empirical results. The findings support 12 of the 13 proposed hypotheses, with a focus on hypotheses that directly and indirectly influence behavioral intention, as well as one hypothesis that was not supported by the data.

### Findings of the direct hypotheses

H3: Results indicated that perceived usefulness positively influenced behavioral intention, with a path coefficient of 0.170. This finding supports the hypothesized relationship in the TAM framework and aligns with the conclusions of Song et al. [13], Wei et al. [26] , and Pal et al. [47]. This study highlights the fact that increasing older adults' perceived usefulness of smart home systems is crucial for enhancing their behavioral intentions. Therefore, during the development and design of smart home systems, it is essential to focus on improving system functionality and practicality to strengthen the behavioral intentions of elderly users.

H6: The study found that facilitating conditions positively influenced behavioral intention, with a path coefficient of 0.195. This conclusion is supported by the findings of Zhong et al. [29], Maswadi et al. [30], and Jahanshahi et al. [66]. Elderly users often experience greater technological anxiety than do younger individuals when confronted with new technologies. Providing facilitating conditions, such as technical training and guidance, can effectively alleviate anxiety and enhance behavioral intention. Hence, during the initial promotion of smart home systems, it is essential to provide facilitating conditions for the elderly.

**Table 9. Model path analysis results.**

| Hypothesis | Path | Standardized coefficient (β) | t-Statistics | p-Value | Hypothesis status |
|---|---|---|---|---|---|
| H1 | PEOU→PU | 0.119 | 2.330 | 0.020 | ✓ |
| H2 | PEOU→BI | 0.103 | 2.055 | 0.040 | ✓ |
| H3 | PU→BI | 0.170 | 3.947 | 0.000 | ✓ |
| H4 | FC→PU | 0.108 | 2.238 | 0.025 | ✓ |
| H5 | FC→PEOU | 0.312 | 8.044 | 0.000 | ✓ |
| H6 | FC→BI | 0.195 | 4.686 | 0.000 | ✓ |
| H7 | SI→PU | 0.162 | 3.468 | 0.001 | ✓ |
| H8 | PT→PU | 0.266 | 5.594 | 0.000 | ✓ |
| H9 | PT→BI | 0.026 | 0.568 | 0.570 | ✗ |
| H10 | PC→BI | -0.198 | 4.332 | 0.000 | ✓ |
| H11 | COM→PU | 0.110 | 2.261 | 0.024 | ✓ |
| H12 | COM→PEOU | 0.378 | 9.709 | 0.000 | ✓ |
| H13 | COM→BI | 0.179 | 3.931 | 0.000 | ✓ |

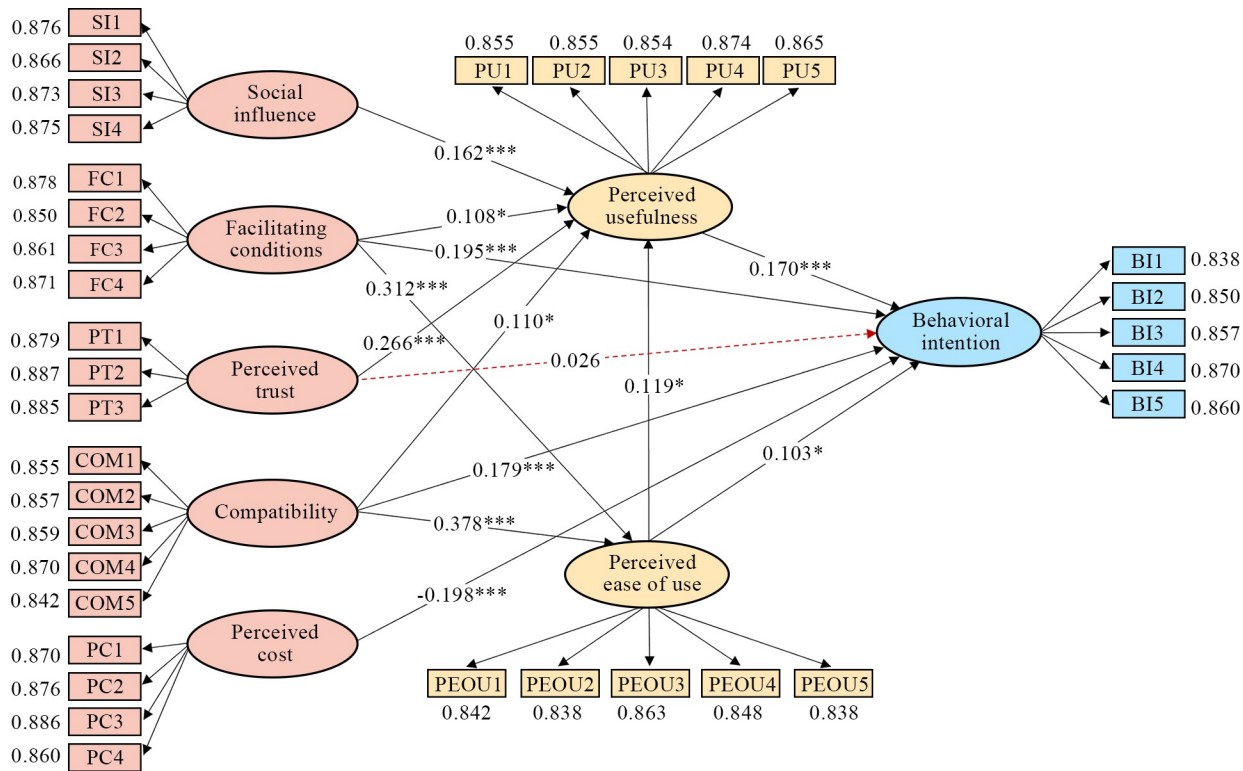

**Fig 3. Model path analysis results. Note**: *p < 0.05, ***p < 0.001. The dashed red line represents an invalid hypothesis.

H10: Results indicated that perceived cost negatively influenced behavioral intention, with a path coefficient of -0.198. This finding supports previous research [58, 59, 73], suggesting that perceived cost can inhibit users' behavioral intentions. High costs may impose a financial burden on users, thereby reducing their intention to engage in smart home systems. Therefore, during the development and early promotion of smart home systems, it is crucial to consider the economic capacity of the elderly to enhance their intentions to use these systems.

H13: Compatibility positively influenced behavioral intention, with a path coefficient of 0.179. This finding was consistent with those reported by Lui et al. [60], Octavius and Antonio [76], and Shetu et al. [102]. This suggests that the compatibility between smart home systems and the habits, experiences, and existing devices of the elderly can reduce technological barriers and enhance behavioral intentions toward these systems. Thus, during the development and design of smart home systems, developers and designers should prioritize system compatibility.

### Findings of the mediating hypotheses

H5: Facilitating conditions positively influenced perceived ease of use, which in turn affects behavioral intention. This finding aligns with Chen and Aklikokou's [64] study, suggesting that perceived ease of use mediates the relationship between facilitating conditions and behavioral intention. The indirect influence of facilitating conditions provides a more comprehensive understanding of older adults' behavioral intentions. Compared with direct

effects, indirect effects are more suitable for long-term promotion and user engagement. In the long-term promotion of smart home systems, developers and policymakers should provide appropriate technical training and guidance to elderly users.

H7: Social influence positively affects perceived usefulness, which in turn affects behavioral intention. This finding is consistent with previous studies [44, 56]. When others in the social networks of the elderly hold positive attitudes and perceptions toward smart home systems, the perceived usefulness of the system increases, thereby enhancing their intentions to use it. In long-term promotions, developers and designers should recognize the importance of social influence and establish mechanisms for positive user feedback.

H8: Perceived trust positively influences behavioral intention by enhancing perceived usefulness. This finding was consistent with those of previous studies [57, 68]. Dhagarra et al. [57] noted that the effect of perceived trust on behavioral intention is mediated by perceived usefulness. Given user concerns about data security and privacy protection [103] and the trust issues that arise from these concerns, developers and designers should continuously improve technologies to ensure data security and privacy of elderly users. In the long term, this helps enhance the perceived usefulness and ultimately have a positive effect on behavioral intention.

H12: Compatibility positively influences perceived ease of use, which in turn affects behavioral intention. This finding was consistent with those reported by Yan and Lee [27] and Al-Bashayreh et al. [55]. These results further validate the mediating effect of perceived ease of use. Compatibility reduces the time and effort required to use the system, thereby enhancing elderly users' perceived ease of use and ultimately strengthening their intention to use the system. Therefore, compatibility is a key consideration for developers and designers in user development.

### Findings of the invalid hypothesis

H9: The hypothesized relationship between perceived trust and behavioral intention is not supported. This result contradicts with those of previous studies [58, 67, 70]. However, perceived trust indirectly influenced behavioral intention through perceived usefulness. A possible explanation for this phenomenon is that elderly users may prioritize the system's practical benefits over a mere sense of trust when considering smart home systems. This finding suggests that, in the development of smart home systems, greater emphasis should be placed on enhancing their functionality and practicality.

## Conclusion

This study identified the direct and indirect factors influencing the behavioral intentions of the elderly toward smart home systems and provided an in-depth analysis of the complex mechanisms behind these influences. The findings not only deepen our understanding of older adults' acceptance of smart home systems but also offer valuable practical guidance for developers, designers, and policymakers.

### Theoretical contributions

This study examines the key factors influencing older adults' intentions to use smart home systems, deepens our understanding of their behavioral intentions, and expands the theoretical framework of technology acceptance among the elderly.

First, by integrating the TAM, UTAUT, and their extended variables, this study developed a comprehensive model that revealed the complex mechanisms driving older adults' behavioral intentions toward smart home systems. The findings demonstrate that perceived usefulness, facilitating conditions, compatibility, and perceived cost have significant direct effects on behavioral intention. This validates the applicability of the TAM and UTAUT in this context and underscores the importance of the extended variables (compatibility and perceived cost) in influencing behavioral intention, thereby enriching the existing theoretical framework.

In addition, this study explored the mediating effects of perceived ease of use and perceived usefulness. The results demonstrated that facilitating conditions and compatibility indirectly influenced behavioral intention through perceived ease of use, whereas social influence and perceived trust indirectly affected behavioral intention through perceived usefulness. These findings revealed the complex mechanisms underlying the formation of behavioral intentions among the elderly, offering new perspectives for understanding their intentions toward smart home systems and expanding the theoretical model.

## Practical implications

This study clarified the distinct mechanisms by which direct and indirect factors influenced behavioral intentions among the elderly. Direct influencing factors have a straightforward relationship with behavioral intention, making them suitable for scenarios in which a short-term effect is required in the early stages of product promotion. By contrast, indirect influencing factors affect behavioral intention through mediating variables, making them more applicable to long-term promotion and user development strategies. The practical implications of these findings are explored from the perspectives of the elderly, developers, designers, and policymakers.

**For the elderly.**   These findings indicated that the elderly prioritized perceived usefulness, compatibility, facilitating conditions, and perceived cost when selecting smart home systems. Based on these insights, we can better understand the needs of elderly users and provide them with tailored smart home solutions, ultimately enhancing their quality of life and independence.

**For developers and designers.**   In the development and design of smart home systems, developers and designers should focus on factors such as perceived usefulness, compatibility, and perceived cost that directly influence the interests of elderly users. To enhance the perceived usefulness of the system, attention should be paid to functionality, particularly in improving safety and health management features, so that elderly users can experience tangible benefits in their daily lives. Ensuring compatibility requires a close alignment of the system with the daily habits, experiences, and existing devices of elderly users, thereby minimizing technological barriers. In addition, cost is a critical concern for elderly users. Controlling system costs during the development and promotion stages and offering flexible payment options can alleviate the financial burden on elderly users.

The practical significance of the indirect influencing factors lies in their long-term impact on the behavioral intentions of the elderly. In the long-term promotion of smart home systems, developers and designers should consider the indirect effects of social influence on the behavioral intentions of older adults. This can be achieved by actively collecting feedback and suggestions from elderly users regarding smart home systems and promptly addressing and resolving their concerns. Establishing a proactive user feedback mechanism is essential for continuously improving and optimizing the user experience, which in turn enhances users' willingness to use the system. Therefore, developers and designers should not overlook the indirect effects of perceived trust. The use of smart home systems must be grounded in robust

data security and privacy protection. Developers should continuously optimize technology and services to ensure the security of user information and prevent trust crises arising from data breaches.

**For policymakers.** In the early stages of promoting smart home systems, policymakers should emphasize the direct influence of facilitating conditions on the behavioral intentions of the elderly. Creating smart home system experience zones in communities through community services or partner organizations can allow the elderly to personally experience the functionality and convenience of these systems. Policymakers could also offer free or low-cost technical training to help older adults overcome technology-related anxiety. Policymakers should also consider the indirect influence of perceived cost on behavioral intention. In the long-term promotion of smart home systems, collaborating with the government or community organizations to introduce subsidies or discount programs can effectively reduce the financial burden on the elderly, thereby increasing their willingness to use these systems.

## Research limitations and future research

Although this study provides important insights into the behavioral intentions of the elderly toward smart home systems, it has some noteworthy limitations.

First, the sample was drawn primarily from the relatively economically developed regions of China's eastern coastal areas. This limits the generalizability of our findings. Future research should consider expanding the sample to include a broader geographic range to obtain a more comprehensive understanding of the behavioral intentions of the elderly toward smart home systems across different regions.

Moreover, the study employed a cross-sectional design, which limited its ability to capture the evolution of older adults' behavioral intentions over time. Therefore, future research should consider using a longitudinal approach to dynamically track changes in older adults' attitudes toward smart home systems, thereby providing deeper insights into long-term trends and the underlying factors that influence the behavioral intention.

Furthermore, this study primarily relied on quantitative research methods, which offered advantages such as objectivity of data, reproducibility of results, and the ability to handle large samples. However, quantitative research may not fully capture the intrinsic motivations of older adults in their social and cultural contexts. Future research could address this limitation by incorporating qualitative methods such as in-depth interviews or focus group discussions to complement the quantitative approach and provide more comprehensive insights.

Finally, this study focused on smart home systems as a whole and did not differentiate between specific types of systems, such as smart security, smart lighting, and smart climate control systems. Distinct factors may influence different types of smart home systems. Therefore, future research should separately investigate the factors influencing these systems to provide more precise guidance for stakeholders.

## Supporting information

**S1 File. Inclusivity in global research.**
(PDF)

## Acknowledgments

We sincerely thank the Biomedical Ethics Committee of Qufu Normal University for their ethical review and approval of this study. We also express our deep appreciation to all the respondents who participated in the pilot study; their valuable feedback was instrumental in refining

our research design. Additionally, we are especially grateful to the experts who contributed to this study; their professional advice on the questionnaire design was crucial to the successful completion of this research.

## Author Contributions

**Conceptualization:** Yuan Wang, Norazmawati Md. Sani.

**Data curation:** Yuan Wang, Yinhong Hua.

**Formal analysis:** Yuan Wang, Yinhong Hua.

**Investigation:** Yuan Wang.

**Methodology:** Yuan Wang.

**Resources:** Yuan Wang.

**Software:** Yuan Wang.

**Supervision:** Norazmawati Md. Sani, Yinhong Hua.

**Validation:** Yuan Wang.

**Visualization:** Yuan Wang.

**Writing – original draft:** Yuan Wang.

**Writing – review & editing:** Yuan Wang, Norazmawati Md. Sani, Yinhong Hua, Qianling Jiang, Long Zhao.

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
