## [Decision Letter · Decision Letter 0]

12 Jul 2024

PONE-D-24-20607Factors Affecting Elderly Behavioral Intention Towards Smart Home Systems: A Cross-Sectional Study from China's Eastern CoastPLOS ONE

Dear Dr. Wang,

Thank you for submitting your manuscript to PLOS ONE. After careful consideration, we feel that it has merit but does not fully meet PLOS ONE’s publication criteria as it currently stands. Therefore, we invite you to submit a revised version of the manuscript that addresses the points raised during the review process.

We look forward to receiving your revised manuscript.

Kind regards,

Mohammed A. Al-Sharafi

Academic Editor

PLOS ONE

4. In the online submission form, you indicated that [To protect the privacy of the respondents, the data supporting the findings of this study are available upon request from the corresponding author.]. 

Editor Comments:

1. The introduction must be enhanced by a clearer statement of the research gap and the study's specific objectives.

2. In the introduction section, it could be beneficial to provide more specific statistics or references related to the elderly population in China's eastern coastal regions.

3. The integration of TAM and UTAUT in your research model needs stronger justification. The substitution of performance expectancy and effort expectancy with perceived usefulness and perceived ease of use is not clearly explained. These constructs are conceptually similar, so explain why the TAM constructs offer unique advantages. Discuss the theoretical and practical considerations for using specific constructs from each model. Include any pilot study results that support your choice.

4. The hypotheses section is currently shallow and lacks clarity. It would benefit from a more structured and detailed presentation. Each hypothesis should be separately stated and explicitly linked to the theoretical constructs, supported by recent references from the last 3 years.

5. Subsection 3.3 Model Construction is not needed. Consider providing a visual model of the theoretical framework (Figure 1) early in the section to guide readers before the hypotheses are developed.

6. In the Methodology section, the description of the sample size calculation can be streamlined for better clarity and conciseness. Additionally, please provide more detailed information on the pilot study, including the number of participants, the feedback received, and how the suggestions from domain experts were incorporated into the final questionnaire design.

7. Add a new subsection within the Methodology section to justify using PLS-SEM in your study. This subsection should elaborate on why PLS-SEM is the most appropriate analytical technique for your research.

8. In survey research, positional bias can occur, leading to skewed results. It is crucial to discuss the procedures used to mitigate this bias within your study. Please add a new subsection in the analysis section to address this issue.

9. The discussion could be more focused. Focus on the most critical results and their implications. Differentiate between direct and indirect effects and their practical significance.

10. The theoretical contributions and practical implications sections require further development. In the theoretical contributions, there is a need for a deeper exploration of how your findings specifically contribute to these theories. Regarding the practical implications, please provide actionable recommendations for smart home system developers, designers, policymakers, and other stakeholders.

11. There are many grammatical errors throughout the manuscript. A thorough proofreading is recommended.

12. Ensure all references are complete and formatted according to the PlosOne guidelines.

Reviewers' comments:

Reviewer's Responses to Questions

**Comments to the Author**

1. Is the manuscript technically sound, and do the data support the conclusions?

Reviewer #1: Yes

Reviewer #2: Yes

2. Has the statistical analysis been performed appropriately and rigorously? 

Reviewer #1: I Don't Know

Reviewer #2: Yes

3. Have the authors made all data underlying the findings in their manuscript fully available?

Reviewer #1: Yes

Reviewer #2: Yes

4. Is the manuscript presented in an intelligible fashion and written in standard English?

Reviewer #1: No

Reviewer #2: Yes

5. Review Comments to the Author

Reviewer #1: PLOS ONE

Factors Affecting Elderly Behavioral Intention Towards Smart Home Systems: A Cross-Sectional Study from China's Eastern Coast

Reviewed by Muhammed Ibrahim

Introduction:

The introduction needs to be harmonised, there are too many paragraphs.

Related Work:

The objectives should be in the introduction section, not in the literature review.

Theoretical Framework and Research Hypotheses

The authors need to explain the strength and weakness of the chosen theory, and justify why chosen it for the study.

Methodology:

The methodology needs corrections regard which research designed employed and all the section needs to be re structure.

Results:

The result is well presented.

Conclusion:

The conclusion section is well-structured and effectively summarizes the key findings of the study.

General Comments

In general, the authors need to proofread the whole article, and check the grammar.

Reviewer #2: Thanks for giving me a chance to review this manuscript. This is an interesting topic. The author(s) tries to work significantly. This study aims to address this gap by constructing a theoretical model of elderly user behavioral intention, drawing upon the Technology Acceptance Model (TAM) and the Unified Theory of Acceptance and Use of Technology (UTAUT). However, still, some of the anomalies I found during the review process are addressed below, which may help further develop the study. (Comments are included in the attached file.)

6. PLOS authors have the option to publish the peer review history of their article (what does this mean?). If published, this will include your full peer review and any attached files.

Reviewer #1: **Yes: **Muhammed Ibrahim

Reviewer #2: No

---

## [Author Response · Author response to Decision Letter 0]

21 Aug 2024

Dear Editor and Reviewers of PLOS ONE,

We have revised the manuscript based on the suggestions from the editor and reviewers, and we have addressed each of their comments point by point in the "Response to Reviewers" document. While the content has been copied here, due to limitations with displaying images and text formatting, we kindly request that you refer to the "Response to Reviewers" document for the complete and detailed responses.

Response to the Editor’s comments:

Thank you for taking the time to review our manuscript and for your valuable suggestions. In the “Revised Manuscript with Track Changes,” we have highlighted our responses to your comments in gray for your convenience.

1. The introduction must be enhanced by a clearer statement of the research gap and the study's specific objectives.

Our response:

Thanks very much for your nice comments. We agree that the introduction must be enhanced by a clear statement of the research gap and objectives. The modified content is as follows:

“Despite the significant improvements that smart home systems can offer to the elderly [25-28], their adoption rate in this demographic remains relatively low [36, 37]. Therefore, it is essential to explore the key factors influencing the behavioral intentions of older adults. Although existing studies have extensively examined the technical aspects of smart home systems [21, 38, 39], there is a significant lack of empirical research from a user perspective, particularly in the Chinese context. This study aims to investigate the behavioral intentions of the elderly in the eastern coastal regions of China toward smart home systems” (Page 7, Line 134-144).

2. In the introduction section, it could be beneficial to provide more specific statistics or references related to the elderly population in China's eastern coastal regions.

Our response:

Thank you for your valuable suggestion. In response, we have expanded the introduction to include a more detailed description of the aging population in China’s eastern coastal regions, based on the most recent census data. Furthermore, we have created Figure 1, which presents the data for the ten provinces with the largest aging populations in China. Most of these provinces are concentrated in the eastern coastal regions. The modified content is as follows：

“According to the ‘Bulletin of the Seventh National Population Census,’ China has the world’s largest elderly population, particularly concentrated in the eastern coastal provinces, such as Shandong, Jiangsu, Guangdong, Hebei, and Zhejiang [4]. Fig 1 A presents the top 10 provinces by elderly population, and Fig 1 B illustrates their geographical locations” (Page 4, Line 79- Page 5, Line 83).

Fig 1. The top 10 provinces in China with the largest elderly populations in 2020.

(A) Number of elderly people in the top 10 provinces, (B) The geographical locations of these provinces (Page 5, Line 97- 100).

3. (1) The integration of TAM and UTAUT in your research model needs stronger justification. Discuss the theoretical and practical considerations for using specific constructs from each model. Include any pilot study results that support your choice.

Our response:

 Thank you for your insightful comments, which we fully agree with. Our literature review confirmed that TAM and UTAUT are the most widely adopted theoretical models in this field of research. Specifically, the variables of TAM (perceived usefulness and perceived ease of use) and UTAUT (social influence and facilitating conditions) are frequently applied in studies examining the acceptance of smart home technology among the elderly. Moreover, we selected TAM and UTAUT because perceived usefulness and perceived ease of use have moderating effects between the UTAUT variables, extended variables, and behavioral intention. The modified content is as follows：

“In the context of smart home systems for the elderly, Pal et al. [55] examined the factors influencing the adoption of smart home services by the elderly using the TAM. Etemad-Sajadi and Gomes Dos Santos [56] explored the use of IoT health technologies in the homes of older adults guided by the TAM. Yan & Lee [57] developed a model for a smart home healthcare system for the elderly, grounded in based on the TAM. These studies demonstrate that perceived usefulness and perceived ease of use have strong explanatory power for the elderly’s adoption of smart home technologies and devices. Although multiple studies indicate that the TAM is applicable in the context of smart homes for the elderly, scholars have pointed out that it overlooks the influence of social factors on the technology adoption process [65]. This limitation could be overcome by incorporating external variables” (Page 14, Line 265- Page 15, Line 285)

“Notably, the UTAUT includes social influence, thus addressing the limitations of the TAM, which tends to overlook social factors” (Page 16, Line 304- 306).

“The UTAUT has also been widely used in studies related to smart homes for older adults. Scholars have pointed out that the UTAUT model was initially developed based on systems in workplace environments, in which the determining factors may differ from those in a home setting [65]. However, many empirical studies on UTAUT have been conducted in home environments [44, 45, 50, 54, 55, 59-61]. Arar et al. [59] used the UTAUT framework to analyze the acceptance and preferences of older adults in Dubai, UAE, toward smart home technology. Maswadi et al. [61] employed the UTAUT model to study the behavioral intentions of the elderly in Saudi Arabia to adopt smart home technologies. Zhong et al. [50] used the UTAUT framework to investigate differences in the acceptance of smart home voice assistants across age groups. These studies largely identified social influence and facilitating conditions as key variables and validated their effectiveness” (Page 16, Line 310- Page 17, Line 326).

“Several scholars have integrated variables from the TAM and UTAUT for empirical research [44, 53, 58]. Researchers have combined variables such as social influence, facilitating conditions, perceived usefulness, perceived ease of use, and compatibility to construct a comprehensive model examining older adults’ behavioral intentions toward smart wearable devices [44]. The results indicated that perceived usefulness and perceived ease of use mediated the relationship between various factors and behavioral intention, thus enhancing the understanding of user behavioral intentions” (Page 18, Line 363- Page 19, Line 369).

(2) The substitution of performance expectancy and effort expectancy with perceived usefulness and perceived ease of use is not clearly explained. These constructs are conceptually similar, so explain why the TAM constructs offer unique advantages.

Our response:

 Thank you for highlighting this point. We have addressed it by adding a subsection titled “A Comparison between the TAM and UTAUT” within the “Literature Review” section to provide a detailed explanation. The details are as follows:

“A comparison between the TAM and UTAUT shows that the perceived usefulness in the TAM corresponds to performance expectancy in the UTAUT, whereas perceived ease of use corresponds to effort expectancy [61, 81]. Perceived usefulness and perceived ease of use were selected as the research variables for two main reasons. First, compared with performance expectancy and effort expectancy, perceived usefulness and perceived ease of use have been widely applied in studies on the acceptance of smart home technologies, confirming their applicability in the smart home domain [10, 23, 46, 57]. Second, multiple studies have confirmed the mediating effects of perceived usefulness and perceived ease of use on other research variables and behavioral intention [82-84]. The mediating roles of perceived usefulness and perceived ease of use allow for a more comprehensive understanding of the complex mechanisms underlying the behavioral intentions of the elderly” (Page 17, Line 329- Page 18, Line 345).

4. The hypotheses section is currently shallow and lacks clarity. It would benefit from a more structured and detailed presentation. Each hypothesis should be separately stated and explicitly linked to the theoretical constructs, supported by recent references from the last 3 years.

Our response:

Thank you for your detailed suggestions regarding the hypothesis section. We have thoroughly revised it to improve clarity. To enhance the structure, we have introduced subheadings in the “Hypotheses Development” section, specifically “Variables of the TAM,” “Variables of the UTAUT,” and “Extended Variables.” Under these subheadings, we have elaborated on the research variables and linked them to their corresponding theoretical constructs. Additionally, we have separated the previously grouped hypotheses, presenting each one individually. Given the length of this section, we have not included it here. However, in the “Revised Manuscript with Track Changes,” these modifications are highlighted in gray for your review (Page 22, Line 426- Page 32, Line 641).

We have also updated this section with the latest literature. In the “Revised Manuscript with Track Changes,” the new and removed references are marked in different colors in the “Reference” section for clarity. New references, numbered 68-77, 79-80, 82-84, 90-92, 94-97, and 99-103, 106-109, are highlighted in blue. Additionally, we have removed some older references, numbered 85-87, 89, 98, 104, and 105, which are marked in red (Page 82, Line 1637- Page 85, Line 1787).

5. Subsection 3.3 Model Construction is not needed. Consider providing a visual model of the theoretical framework (Figure 1) early in the section to guide readers before the hypotheses are developed.

Our response:

Thanks very much for your nice comments. We have removed “Subsection 3.3 Model Construction” (Page 32, Line 643- Page 33, Line 666) and integrated a visual model of the theoretical framework into the earlier “Theoretical background” section (Page 20, Line 392-394). Consequently, the numbering of the figures has been updated, with the new image now labeled as “Fig 2,” as detailed below.

Fig 2. Theoretical framework.

6. (1) In the Methodology section, the description of the sample size calculation can be streamlined for better clarity and conciseness. (2) Additionally, please provide more detailed information on the pilot study, including the number of participants, the feedback received, and how the suggestions from domain experts were incorporated into the final questionnaire design.

Our response:

Thank you for your comments. We fully agree with your suggestions and have revised the “Sample Size” section to enhance clarity and brevity. Due to the length of this section, the details are not listed here. Please refer to the “Sample Size” section in the “Revised Manuscript with Track Changes” for a comprehensive view of the modifications (Page 39, Line 737- Page 40, Line 762).

(2) Additionally, please provide more detailed information on the pilot study, including the number of participants, the feedback received, and how the suggestions from domain experts were incorporated into the final questionnaire design.

Our response:

We appreciate your suggestion and fully agree with it. We’ve also included a “Pilot Study” section that details the number of participants, the feedback received, and how the experts’ suggestions were integrated into the final questionnaire design. The modified content is as follows:

“A pilot study and expert validation were conducted before collecting data [61]. In this pilot study, 35 participants from diverse backgrounds were invited to evaluate the reliability and representativeness of the questionnaire. After completing the questionnaire, the respondents provided feedback on content relevance, language clarity, and comprehension difficulty. In addition, three experts in smart home technology were consulted to review the questionnaires. They provided in-depth suggestions regarding the structure and content, including the accuracy of wording, professionalism of the language, and optimization of the question order. Based on the feedback from both experts and participants, comprehensive adjustments were made to the questionnaire to ensure its suitability for subsequent data collection” (Page 38, Line 712- 722).

7. Add a new subsection within the Methodology section to justify using PLS-SEM in your study. This subsection should elaborate on why PLS-SEM is the most appropriate analytical technique for your research.

Our response:

Thank you for your valuable feedback, which has been instrumental in improving our manuscript. In response, we have incorporated a new subsection titled “PLS-SEM” within the “Research Methodology” section, in which we have provided a detailed explanation of our rationale for selecting PLS-SEM. The revised content is presented below:

“This study used PLS-SEM for data analysis based on three key considerations. First, compared with CB-SEM, PLS-SEM is more suitable for non-normally distributed data. Because data in this study did not follow a normal distribution, the selection of PLS-SEM ensured the robustness of the analysis and predictive capability of the model [111]. Second, PLS-SEM emphasizes the predictive power and explanatory ability of the model, making it particularly well-suited for models involving multiple variables and complex relationships [112]. As the model in this study involved complex relationships between several variables, PLS-SEM was considered more appropriate. Additionally, PLS-SEM has been widely applied in this field, providing a reference and guidance for our methodological choice [46, 57, 61]” (Page 34, Line 680- 689).

8. In survey research, positional bias can occur, leading to skewed results. It is crucial to discuss the procedures used to mitigate this bias within your study. Please add a new subsection in the analysis section to address this issue.

Our response:

Thank you for your insightful advice. In response, we have added a subsection titled “Mitigation of Positional Bias” in the “Data Analysis and Results” section to address this issue. The revised content is provided below:

“Position bias refers to a phenomenon in which user behavior and selection outcomes are influenced by the positions of items or options on a page or list [122]. To address this issue, a small-scale pilot study was conducted by randomizing the order of the questions and options in the questionnaire to assess potential position bias. The order of the questions and scale items was adjusted based on participant feedback and expert recommendations, and neutral wording was used to minimize language bias. These steps were taken to reduce the impact of position bias on the study results” (Page 46, Line 865- Page 47, 872).

9. The discussion could be more focused. Focus on the most critical results and their implications. Differentiate between direct and indirect effects and their practical significance.

Our response:

Thank you for your valuable suggestion, which we have taken into careful consideration. In response, we have thoroughly reorganized and revised the “Discussion” section, addressing key hypotheses within the newly structured “Findings of the Direct Hypotheses” and “Findings of the Mediating Hypotheses” sections. Due to the length of the content, we are unable to list it here, but you can find the detailed revisions in the “Discussion” section of the “Revised Manuscript with Track Changes” (Page 54, Line 976- Page 62, Line 1131).

Additionally, we have revised the descriptions of the direct and indirect effects, as well as their practical implications, in the “Conclusion” section as follows：

“This study clarified the distinct mechanisms by which direct and indirect factors influenced behavioral intentions among the elderly. Direct influencing factors have a straightforward relationship with behavioral intention, making them suitable for scenarios in which a short-term effect is required in the early stages of product promotion. By contrast, indirect influencing factors affect behavioral intention through mediating variables, making them more applicable to long-term promotion and user development strategies. The practical implicatio

---

## [Decision Letter · Decision Letter 1]

17 Sep 2024

Factors affecting the elderly's behavioral intention toward smart home systems: A cross-sectional study from China's eastern coast

PONE-D-24-20607R1

Dear Dr. Wang,

We’re pleased to inform you that your manuscript has been judged scientifically suitable for publication and will be formally accepted for publication once it meets all outstanding technical requirements.

Kind regards,

Mohammed A. Al-Sharafi

Academic Editor

PLOS ONE

Reviewers' comments:

Reviewer's Responses to Questions

**Comments to the Author**

1. If the authors have adequately addressed your comments raised in a previous round of review and you feel that this manuscript is now acceptable for publication, you may indicate that here to bypass the “Comments to the Author” section, enter your conflict of interest statement in the “Confidential to Editor” section, and submit your "Accept" recommendation.

Reviewer #1: All comments have been addressed

Reviewer #2: All comments have been addressed

2. Is the manuscript technically sound, and do the data support the conclusions?

Reviewer #1: (No Response)

Reviewer #2: Yes

3. Has the statistical analysis been performed appropriately and rigorously? 

Reviewer #1: (No Response)

Reviewer #2: Yes

4. Have the authors made all data underlying the findings in their manuscript fully available?

Reviewer #1: (No Response)

Reviewer #2: Yes

5. Is the manuscript presented in an intelligible fashion and written in standard English?

Reviewer #1: (No Response)

Reviewer #2: Yes

6. Review Comments to the Author

Reviewer #1: (No Response)

Reviewer #2: (No Response)

7. PLOS authors have the option to publish the peer review history of their article (what does this mean?). If published, this will include your full peer review and any attached files.

Reviewer #1: **Yes: **Muhammed Ibrahim

Reviewer #2: No

---

## [Editor Report · Acceptance letter]

23 Sep 2024

PONE-D-24-20607R1 

PLOS ONE

Dear Dr. Wang, 

I'm pleased to inform you that your manuscript has been deemed suitable for publication in PLOS ONE. Congratulations! Your manuscript is now being handed over to our production team.

Kind regards, 

on behalf of

Dr. Mohammed A. Al-Sharafi 

Academic Editor

PLOS ONE